# Genomic Divergence in Swedish Warmblood Horses Selected for Equestrian Disciplines

**DOI:** 10.3390/genes10120976

**Published:** 2019-11-27

**Authors:** Michela Ablondi, Susanne Eriksson, Sasha Tetu, Alberto Sabbioni, Åsa Viklund, Sofia Mikko

**Affiliations:** 1Department of Veterinary Science, University of Parma, 43126 Parma, Italy; michela.ablondi@unipr.it (M.A.); alberto.sabbioni@unipr.it (A.S.); 2Department of Animal Breeding and Genetics, Swedish University of Agricultural Sciences, PO Box 7023, S-75007 Uppsala, Sweden; susanne.eriksson@slu.se (S.E.); sasha.tetu.etu@univ-lille.fr (S.T.); asa.viklund@slu.se (Å.V.)

**Keywords:** Swedish Warmblood, horse, population structure, show jumping, signatures of selection

## Abstract

The equestrian sport horse Swedish Warmblood (SWB) originates from versatile cavalry horses. Most modern SWB breeders have specialized their breeding either towards show jumping or dressage disciplines. The aim of this study was to explore the genomic structure of SWB horses to evaluate the presence of genomic subpopulations, and to search for signatures of selection in subgroups of SWB with high or low breeding values (EBVs) for show jumping. We analyzed high density genotype information from 380 SWB horses born in the period 2010–2011, and used Principal Coordinates Analysis and Discriminant Analysis of Principal Components to detect population stratification. Fixation index and Cross Population Extended Haplotype Homozygosity scores were used to scan the genome for potential signatures of selection. In accordance with current breeding practice, this study highlights the development of two separate breed subpopulations with putative signatures of selection in eleven chromosomes. These regions involve genes with known function in, e.g., mentality, endogenous reward system, development of connective tissues and muscles, motor control, body growth and development. This study shows genetic divergence, due to specialization towards different disciplines in SWB horses. This latter evidence can be of interest for SWB and other horse studbooks encountering specialized breeding.

## 1. Introduction

Throughout history, horses have played many roles in serving humans. In modern society, horses are selectively bred to suit mainly leisure and sport activities. Generally, selection leads to a reduction in genetic diversity at genome level, primarily in genomic regions where genes affecting the desired phenotypes are located. If animals of the same breed are selected for different purposes, different breeding goals may lead to genetic subpopulations [1]. By studying patterns of genetic variation in the genome, we can learn more about the population’s structure and consequences of selection.

The Swedish Warmblood (SWB) originated from cavalry horses in the 18th century, and the studbook was founded in 1928 with the initial aim to breed versatile horses suited for multiple equestrian purposes [2]. In the last decades, the increased demand for highly competitive horses has motivated SWB breeders to focus their breeding on either show jumping or dressage, which are the two main equestrian disciplines in Sweden. Consequently, the SWB breeding program has evolved towards aiming at genetic improvement for specialized disciplines. Since 2002, breeding stallions are selected as either dressage or show jumping stallions [3], and, since 2006, the SWB studbook has published one main estimated breeding value (EBV) for show jumping and one for dressage performance based on scores at young horse tests and competition data [4]. Yet, all SWB horses tested at young horse evaluations and stallion performance tests are assessed for both gait and show jumping traits, regardless of the discipline they are bred for. The genetic trend based on EBVs has shown that genetic progress increased substantially in the mid 1980s for both disciplines, and has been considerably higher for show jumping than for dressage, mainly due to successful stallion selection and higher heritability for show jumping traits [4].

While the specialization for different disciplines within a breed implies some challenges for traditional genetic evaluation, it provides possibilities to compare genomic information from horses selected for different purposes. Such knowledge can, in turn, help sharpen genetic selection tools for the future. Recent examples of genomic studies have shown the benefits of studying genetic subpopulations. In cattle and in horses, genetic stratification within breeds, resulting from selection for different purposes, was shown by using Principal Component Analysis (PCA) and Discriminant Analysis of Principal Component (DAPC) [5,6,7]. Petersen et al., (2014) used fixation index (*F_ST_*), PCA and haplotype analyses, to find genomic differences across six performance groups of Quarter Horse, caused by increased specialization over the past 75 years [8]. The presence of signatures of selection within Quarter Horse subpopulations was confirmed in a more recent study [9]. Principal Coordinates Analysis (PCoA) was used to both unravel the origin of the feral horses in Theodore Roosevelt National Park, and to visualize genetic distances among horses belonging to an emergent breed in the Azores [10,11].

Several studies on signatures of selection and population structure in horses based on high-density genotype data have been presented recently [9,12,13,14,15,16]. The degree of homozygosity at haplotype level is useful to detect signatures of positive selection [17]. The Cross Population Extended Haplotype Homozygosity (XPEHH) method estimates the length of extended haplotypes and evaluates differences between two populations, or subpopulations, and it has previously been used to detect signatures of selection in Shetland ponies [18], Thoroughbreds [19], and SWB [16]. Such chromosomal regions, which show signatures of selection, can be large, and comprise many genes as well as regulatory elements [12,13,20,21,22].

In a recent study [16], we found genomic signatures of selection for performance in SWB horses by comparing them to the Exmoor pony breed, which is a breed not selected for sport performance. Our results suggested that genes related to behavior, physical abilities and fertility have been general targets of selection in the SWB breed, regardless of what discipline they were bred for. In our present study we focus on potential differences at genome level within subgroups of the SWB breed to further unravel specific genomic regions under selection for different sport purposes. The aim of this study was therefore to (1) explore the genomic structure in the SWB breed and (2) to detect putative genomic subpopulations of SWB horses based on their EBVs for show jumping. Two methods, PCoA and DAPC, were used to detect population stratification within the SWB breed, while *F_ST_* and XPEHH scanned the genome for potential signatures of selection. To explain the biological importance of selection footprints, we performed functional classification of the genes identified within regions potentially under selection. Our research provides novel insight into how discipline specialization within the SWB breed has shaped the horses’ genomes and offers useful knowledge for forthcoming horse breeding schemes.

## 2. Materials and Methods

### 2.1. Definition of SWB Subpopulations

In this study, we analyzed high-density genotype information from 380 Swedish Warmblood horses born in 2010–2011 (selected tested population, STP). The horses (182 males and 198 females) were assessed in young horse evaluation tests at the age of three. They descended from 145 sires with 1–11 offspring each, and 372 mares with 1–2 offspring each in the study. The proportion of Thoroughbred contribution was calculated based on four generations of each individual pedigree. Breeding values from the latest routine genetic evaluation (2018), estimated in a multi-trait animal model, and based on young horse tests, together with competition data [4], were available for all studied horses. A breeding value equal to 100 denotes the average for all tested horses between four and eighteen years of age in the SWB population. In this study, horses with EBVs for show jumping performance above 100 were classified as showjumping horses (SJ), and horses with EBVs less than 100 as non-showjumping horses (NS) (Figure 1). The majority, but not all, of the NS horses could be described as horses bred for the dressage discipline. Because some degree of preselection of horses shown at young horse tests can be expected, a comparison was made to assess if the 380 horses were representative of the SWB cohort at population level. We tested the equality of mean EBV for show jumping between the STP, and all 1540 horses tested the same years (2013–2014) (total tested population, TTP), as well as all 8273 horses born in the same year’s cohort (reference population, RP). Likewise, we also tested equality of mean EBV between the two subpopulations of TTP horses (SJ and NS) in SAS 9.4. [23].

### 2.2. DNA Isolation, Genotyping and Quality Control

DNA was prepared from 20 hair roots, cut into 5% Chelex 100 Resin (Bio-Rad Laboratories, Hercules, CA, USA), and 1.4 mg/mL Proteinase K (Merck KgaA, Darmstadt, Germany) with a total volume of 200 µL. The samples were vortexed at 1500 rpm for 2 h in 56 °C, followed by heat inactivation of Proteinase K at 96 °C for 10 min. DNA concentration was measured using the Quant-iT^TM^ PicoGreen^TM^ dsDNA Assay Kit (Thermo Fisher Scientific, Santa Clara, CA, USA), and normalized. The DNA was re-suspended in low-TE (1 mM Tris, 0.1 mM EDTA) at a concentration of 10 ng/µL.

All samples were genotyped using the 670 K Affymetrix^®^ Axiom^®^ Equine Genotyping Array (Thermo Fisher Scientific, Santa Clara, CA, USA) [24]. The obtained genotypes of markers included in the 670 K Single Nucleotide Polymorphism (SNP)-chip were remapped from the former reference genome EquCab2 to EquCab3 [25] using a Python script, as described in [26]. Only SNPs located on the 31 autosome chromosomes were retrieved and used in this study (606,287 SNPs). Allosomes were not considered because no Y chromosome data were available, and thus allosomes would not enable homozygosity-based analyses in male individuals. The quality control (QC) was performed in PLINK (v1.9) [27] by removing SNPs with a call rate lower than 0.90, MAF < 0.01 and Hardy–Weinberg equilibrium (HWE) deviation with *p* < 0.0001. All individuals had a call rate higher than 0.90. To determine population structure, a linkage disequilibrium (LD) pruning was applied, excluding SNPs if the LD between each pair of SNPs was greater than 0.5 (r_2_ > 0.5) in a window size of 50 SNPs moving 5 SNPs per window [28].

### 2.3. Genomic Structure of the SWB Population

To assess and describe the potential genetic stratification in the STP of SWB horses, we used PCoA and DAPC. In the case of PCoA, the workflow was as follows: (i) Calculation of pairwise genomic distances between the horses were performed in PLINK (v1.9); (ii) The calculated genomic kinship coefficients were transformed to squared Euclidean distances, and the dissimilarities between the subjects within the matrix were captured in n–1 dimensional spaces of n. observations (eigenvectors), via classical multidimensional scaling (MDS) [29] in R [30]; (iii) The proportion of variation captured by each eigenvector was calculated and the two eigenvectors explaining the largest proportion of the variance were plotted in R as a principal component plot.

The DAPC [31] was performed using the *adegenet* package [32] in R, to further evaluate the presence of two subpopulations in the STP of SWB horses. The Bayesian Information Criterion (BIC) analysis was used to determine the number of subpopulations (K) in the STP. The number of principal components (PCs) to retain in the discriminant step in the DAPC was optimized using the cross-validation procedure in *adegenet*, where the dataset was divided into two sets selected by stratified random sampling: a training set (90% of the data) and a validation set (10% of the data). The optimal number of PCs was chosen based on the mean successful assignment of the predefined subpopulations (SJ and NS) and lowest root mean squared errors.

### 2.4. Divergent Selection Between Subpopulations at Genotype Level

The genetic divergence between SJ and NS was verified by the fixation index (*F_ST_*), as defined by Nei (1987) [33]. Measures of centrality and dispersion were used to compare *F_ST_* values for each SNP. Negative *F_ST_* values were set to zero, because negative values have no biological interpretation [34]. Values were interpreted using the qualitative guidelines proposed by Wright [35], where an *F_ST_* value of 0.15–0.25 indicates large differentiation, 0.05–0.15 indicates moderate differentiation, and *F_ST_* < 0.05 indicates little differentiation among populations [36]. SNPs were plotted relative to their physical position within each autosome using a custom-made script in R. SNPs with an *F_ST_* value within the top 0.1% of the *F_ST_* distribution were considered as potential signatures of selection [37]. 

### 2.5. Divergent Selection Between Subpopulations at Haplotype Level

Haplotypes were phased using Shape-it software [38] and filtered using REHH Package in R to (1) discard haplotypes with missing data, (2) keep only fully genotyped markers [39]. Regions homozygous in one population but polymorphic in the other population were highlighted by the comparison of EHH score of the two subpopulations per SNP [17]. The XPEHH score was computed for each autosomal SNP and it was defined and standardized according to [17,39,40]. The XPEHH score was transformed into *p_XPEHH,_* as shown by [40]:(1)$$p_{XPEHH}=-log_{10}\left(1-\;2\;\left|\Phi \left(XPEHH\right)-0.5\right|\right)$$ where Φ(XPEHH) is the Gaussian cumulative distribution function. The $p_{XPEHH}$ can be interpreted as a two-sided *p*-value in a −log10 scale. Negative XPEHH scores indicate that selection happened in SJ horses, while positive values indicate selection in the NS horses [5]. The False Discovery Rate (FDR) adjustment was performed, applying the Benjamini Hochberg method with FDR equal to 0.05 [41].

### 2.6. Candidate Genes

A genomic region was considered as under potential selection if it contained significant SNPs based on both *F_ST_* and XPEHH analyses. We used the Ensembl gene annotations EquCab3 to identify genes residing within regions extending 250 kb up- and downstream of significant SNP. This was done to include potential effects of regulatory changes on loci at some distance, and to reduce the risk of excluding the outermost parts of the selected haplotypes [42]. Functional classification, statistical overrepresentation of biological processes and pathways (GO terms) of candidate genes for each subpopulation were conducted using PANTHER 14.0 (http://pantherdb.org/) [43] on the Equine reference genome EquCab 3.0). Genes were further compared with previously identified QTL regions in the horse QTL database [44].

### 2.7. Ethical Approval

Ethical approval and consent to participate is not applicable as the hair samples from Swedish Warmblood horses were originally collected for parentage testing and stored in the biobank at the Animal Genetics Laboratory, SLU. The Swedish Warmblood Association approved the samples to be used in this research, and no additional samples were collected for this study.

## 3. Results

### 3.1. Genotypic and Phenotypic Population Structure

From the SNPs genotyped on the 670 K SNP-chip, 606,287 autosomal SNPs were retained after remapping to EquCab3. The total genotyping call rate was equal to 0.96 and no single individuals showed a call rate lower than 0.90, leaving all 380 SWB horses, and 456,381 SNPs in the analyses after QC. The LD pruning left 242,396 SNPs to be used in the PCoA and DAPC analyses. No significant differences in the mean show jumping EBV were found from the comparisons between STP and RP and STP and TTP (*p* > 0.80) (Figure 2).

A total of 191 horses had an EBV for show jumping above 100 and were defined as show jumping horses (SJ), whereas the other 189 horses were defined as non-show jumping horses (NS). The average EBV for show jumping was significantly higher in SJ horses (125) compared to NS horses (77) (*p* < 0.001), corresponding to more than two genetic standard deviations (Table 1). Similarly, the average score for jumping technique at the young horse evaluation was significantly higher in SJ (8.3) than in NS horses (5.6) (*p* < 0.001). On the other hand, no significant difference between the two subpopulations was shown for mean test score for gaits, or for the conformation traits “type” and “head–neck–shoulder”, which are traits not specifically related to show jumping performance. Until October 2018, 155 of the horses had competed in show jumping and 91 in dressage, whereas only 18 had competed in both disciplines. A significantly higher proportion of SJ horses (71%) had competed in show jumping classes compared to NS horses (11%) (Table 1).

### 3.2. Population Substructure

The PCoA revealed some genetic stratification among SWB horses, with 5.2% of the total variance explained by the three main eigenvalues. The first two eigenvalues explained 2.63% and 1.29% of the total genetic variation of the data, as visualized in Figure 3. The SJ horses presented a more scattered cluster based on the second component (eigenvalue 2), indicating higher genetic distance and lower relatedness among individuals compared to the NS horses. Eleven horses, all sired by the same father, clustered as outliers in the PCoA. No offspring from this sire clustered among the other 369 individuals.

The BIC analyses based on the genotype data, setting the number of possible subpopulations (K = 1:5), indicated the most likely number of subpopulations was either two or three (Figure 4a). The presence of two main subpopulations was further supported by the DAPC plot (Figure 4b). The cross-validation test for the number of PCs to retain, showed the highest proportion of the successful assignment to the predefined subpopulations (SJ and NS) and lowest root mean squared errors for 100 PCs. The retained 100 PCs explained ~47% of the total variation. The results obtained from the DAPC analysis supported those obtained from PCoA analysis, with most (96%) of the individuals correctly assigned to their predefined SJ and NS subpopulations based on EBVs (Figure 4b). The 23 individuals in the overlapping area of the density DAPC plot were found to have more than twice as high average Thoroughbred proportion (22.4%) in their pedigree, compared with the remaining 357 horses (9.6%) in this study. In the posterior distribution from the DAPC, the same 23 individuals showed less than 80% likelihood of being assigned to their predefined subpopulations (SJ or NS). All genotyped individuals are presented in Appendix A, where each line represents one individual, and the heat color represents their membership probability, provided by DAPC, of being assigned to the predefined subpopulations (SJ or NS).

### 3.3. Genomic Divergence in SWB Subpopulations

The average value of fixation index (*F_ST_*) between SJ and NS horses was 0.015 with a standard deviation of 0.017. A total of 347 SNPs exceeded the 0.1% threshold for significance, corresponding to *F_ST_* value ≥ 0.12, and were considered as putative signs of divergent selection. These SNPs were distributed in all chromosomes, except ECA24 and ECA29. A total of nine SNPs distributed on ECA1, ECA5 and ECA22, had *F_ST_* values higher than 0.20 (Appendix A). The highest number of significant SNPs was obtained for the ECA1 (*n* = 79, with *F_ST_* values ranging from 0.12–0.23); however, when normalizing for the number of SNPs present in the chromosomes, the highest number of outliers was found in ECA22 and ECA25, with 34 and 27 SNPs, respectively (Figure 5).

### 3.4. Signatures of Selection in SWB Subpopulations

In SJ-horses, the XPEHH-analyses detected positive selection in eight genomic regions distributed in seven chromosomes; one region per chromosome on ECA7, ECA8, ECA13, ECA19, ECA22, ECA31, and two regions on ECA12. In NS-horses, positive selection was detected in four genomic regions; one on ECA1, one on ECA23 and two on ECA25 (Figure 6).

The genomic coordinates of the regions under selection, detected by XPEHH in SJ horses, is shown in Table 2. In SJ horses, two significant regions on ECA8 and ECA31 overlapped with known QTLs for body growth and altitude adaptation. A total of six out of eight regions detected by the XPEHH test overlapped with at least one significant SNP from the *F_ST_* analysis (75% concordance between the two tests). In total, 157 genomics elements were found within the six genomic regions under selection in SJ horses: 114 protein coding genes, one pseudogene, 22 lncRNA, nine snoRNA, seven miscRNA, two miRNA and two snRNA (Appendix A).

The PANTHER overrepresentation analysis recognized 74 genes out of the 114 protein coding genes located within the regions under selection in SJ horses. Eight GO terms were overrepresented for biological processes, as well as four pathways in SJ horses (*p* < 0.01) (Table 3). Sixteen of the 74 genes detected by PANTHER analysis were mainly related to development and regulatory functions within the eight overrepresented biological processes, while six genes were involved in overrepresented pathways.

The genomic coordinates of the regions under selection detected by XPEHH in NS horses are shown in Table 4. In total, 46 genomics elements were found within the four genomic regions under selection in NS horses: 31 protein coding genes, 13 lncRNA, one miRNA and one snRNA (Appendix A).

The significant region on ECA1 in NS horses comprised eleven genes and overlapped with a known QTL for free jumping. Five genes were found in the region on ECA23, seven genes on ECA25:5.3–5.8 Mb, and eight genes on ECA25:37.4–37.9 Mb. All significant regions found with the XPEHH test overlapped with at least one significant SNP from the *F_ST_* estimate in NS horses (100% concordance between the two methods).

The PANTHER overrepresentation analysis recognized 21 genes out of 31 protein coding genes found within the regions indicated to be under selection in NS horses. Five GO terms were overrepresented for biological processes, as well as four pathways in NS horses (*p* < 0.01) (Table 5). Among the biological processes and pathways, three genes were overrepresented in each of them.

## 4. Discussion

### 4.1. Horses Selected for This Study

In the SWB studbook, EBVs are available for both showjumping and dressage performance. In the presented study, we used the show jumping EBVs to divide SWB horses in two potentially genetically different groups, for several reasons. The main reasons were the higher heritability [45], and faster genetic progress for the show jumping rather than the dressage trait in SWB horses [4], but also because horses competing in show jumping are more abundant [45]. Additionally, jumping performance traits are exclusive in the selection of show jumping horses, while traits related to conformation and gait are, to some extent, selected in both disciplines. For the above-mentioned reasons, we decided to use the EBV for show jumping to divide the sampled horses in two groups. The bimodal distribution of EBVs facilitated this division. Nevertheless, the mean EBVs for dressage was significantly higher in the NS group compared to the SJ horses. Thus, we expect that most of NS horses in the study were bred mainly for dressage performance, confirming the rationale behind the use of EBVs to divide the horses in this study. The horses were selected from the ones assessed at young horse tests in 2013 and 2014, with the aim of including horses with different pedigrees and performance levels. We cannot completely rule out the possibility that preselection may have influenced our results. However, the comparison between means and distributions of EBV in the sample (STP) and in the whole cohort of horses born in the same years (RP) did not show any obvious signs of preselection.

### 4.2. SWB Breeding Program and Population Substructure

Our results showed genomic substructures in the population in accordance with the breeding practice applied by most breeders. Already after a few decades of specialized breeding, the SWB population shows a clear tendency of two main subpopulations with significantly different EBVs for show jumping. This process has probably been enhanced by the use of stallions imported from European studbooks with a focus on separate disciplines. The first eigenvalue in the PCoA analysis was unable to completely separate the individuals in two clusters, which is expected, as the process of specialization is ongoing. The more scattered cluster, based on the second eigenvalue in the SJ subpopulation, indicated higher genetic distance and lower relatedness between individuals as compared to the NS subpopulation. This may be explained by a larger breeding cohort, and the extensive use of foreign stallions from several studbooks in the breeding of SWB horses for show jumping. The latter agrees with Thorén Hellsten et al. (2009), where they showed that Holstein, Selle Français and KWPN breeds have had an extensive impact on show jumping performance in SWB horses. In contrast, no specific breed was shown to influence the SWB dressage horses, except for Oldenburg in the very last few years of the studied period [46].

The higher percentage of thoroughbred ancestry in some horses may be likely the reason why the DAPC failed to correctly assign those animals to one specific subpopulation. Likewise, the presence of horses with a higher percentage of thoroughbred ancestry may be the reason why the BIC analysis showed a similar likelihood for two and three subpopulations in the data. However, the distribution of EBVs point at two major SWB subpopulations. By adding a very small third subpopulation would, in this study, not contribute much information.

The low average *F_ST_* between the SJ and NS SWB subpopulations is in line with the relatively recent specialization process towards different disciplines. Therefore, even though we could detect genomic divergence within the breed, there is no evidence to claim the presence of two separate breeds.

### 4.3. Genomic Divergence Between SJ and NS SWB Horses

In the past few years, pursuing signatures of selection in the genome has been widely used to understand the genetic mechanism behind phenotypes, and several approaches have been proposed to detect selection footprints [47]. In this study, we employed two representative methods, *F_ST_* and XPEHH, to explore potential signature of selection in show jumping and non-show jumping horses. Both the *F_ST_* and XPEHH tests are based on population differentiation, and they can be considered complementary in time scale. It takes time to reach fixation, especially for complex traits where the desired phenotype results from mutations in a network of genes, rather than from monogenic point mutations. Compared to allele frequency difference, long-range haplotypes persist for relatively shorter periods of time before being broken down by recombination [48]. Signatures of selection at haplotype level can therefore be interpreted as more recent selection compared to *F_ST_* [5,17]. Nevertheless, both *F_ST_* and XPEHH are effective when the traits selected in one subpopulation are not selected in the other subpopulation, resembling the case of show jumping traits.

None of the potentially under selection regions found in SJ or NS horses overlapped with the genomic signature of selections in SWB when we previously compared them with Exmoor ponies [16]. We hypothesize that the genomic regions found in our previous study are of general importance for equestrian sport performance, while genomic regions under selection in only one of the two subpopulations (SJ and NS) of SWB harbor genes of more specific importance for a sport discipline.

All but one of the nine SNPs with *F_ST_* > 0.20 were located within genes or, alternatively, in their 5’ or 3’ untranslated regions, indicating a possible functional role for these genes in SWB horses. It is important to note that these SNPs may not be the causal variants themselves, but in LD with potential causal variants. One of these SNPs was instead located in a region where a 1 Mb long CNV gain was reported in a Quarter horse [49]. In this region, the divergence seen between SJ and NS SWB horses may therefore be due to heterozygosity of an *indel* rather than a higher homozygosity. The remaining eight divergently selected SNPs are located within five genes: *cyclic GMP–dependent protein kinase 1* (*PRKG1*) on ECA1; *NME/NM23 family member 7* (*NME7*), and *leucin-rich-repeat-containing 7* (*LRRC7*) on ECA5; *p21 (RAC1) activated kinase 5* (*PAK5*), and *phospholipase C beta 1* (*PLCB1*) on ECA22. The three genes *PRKG1*, *NME7*, and *PAK5* all encode kinases that phosphorylate proteins in cell signaling pathways. The Ser/Thr kinase encoded by the *PRKG1* gene has a crucial function in smooth muscle activity, and has been associated with thoracic aortic aneurysm in humans [50]. This gene was also reported in a potential selection signature for ocular size, memory and learning in Thoroughbred horses [51]. The kinase encoded by *NME7* catalyzes nucleotides in the cell biosynthesis, and the Ser/Thr kinase encoded by *PAK5* influences the outgrowth of neurons critical in locomotion and cognition [52]. The gene *LRRC7* encodes *densin-180*, an abundant scaffold protein within the excitatory postsynaptic density (PSD) [53]. The gene *PLCB1* plays an important role in signal transduction from G-protein coupled receptors (GPCRs), which have been associated with risk of developing schizophrenia in humans [54,55]. Overall, most of the genes located within the highest divergent regions detected by *F_ST_* are involved in cell signaling cascades in the central nervous system, indicating the potential presence of mental and neuronal differences between horses selected for show jumping or dressage.

Overall, we observed more and longer regions under potential selection in SJ compared to NS horses, which may be a consequence of the more rapid genetic gain in show jumping performance than in non–show jumping performance traits in SWB [4]. It is noteworthy that, within significant regions, we detected both protein-coding genes and non-coding RNAs (ncRNA). Non-coding RNAs are involved in gene regulation by epigenetic modifications, and may regulate other genes in their genomic vicinity [56,57]. Mutations in regulatory elements may be a quicker way for genomes to evolve, compared to coding mutations with high constrains to prevent accumulation of deleterious missense mutations [58].

### 4.4. Selection for High Mobility and Relaxed Locomotion in NS SWB Horses

In NS horses, five genes were overrepresented in the biological processes and pathways analysis; sec61 translocon beta subunit (SEC61B); sphingomyelin synthase 1 (SGMS1); phosphatase and tensin homologue (PTEN); 3′-phosphoadenisine-5′-phosphosulphate synthase 2 (PAPSS2); and transforming growth factor beta receptor 1 (TGFBR1). Three of these genes—SGMS1, PTEN, and PAPSS2—are located within the significant region on ECA1:40,592,555–43,510,660. This ECA1 region was found to be under selection in racing Quarter Horses [9], and overlaps with a known QTL for free jumping in Hanoverian horses [59], suggesting that the homozygous haplotype in the NS SWB horses is unfavorable for jumping performance. The selection pressure on this ECA1 region is further supported by the high divergence detected by F_ST_ between SJ and NS horses, as mentioned above. Mice haplo-insufficient for PTEN show abnormal social behavior and exaggerated reactions to sensory stimuli [60,61], and, in humans, PAPSS2 influences the odds ratio for being an exerciser [62], suggesting the importance of the region for equine sport performance. It was also shown that PAPSS2 mRNA is down-regulated by transforming growth factor beta (TGF-beta) signaling, mediated by its receptor TGFBR1 [63,64]. The equine gene TGFBR1, located on ECA25:5,277,465–5,984,107, was found to be overrepresented in the activin receptor signaling pathway and TGF-beta signaling pathway. Mutations in TGFBR1 have been associated with aortic aneurysms, as well as impaired connective tissues leading to joint laxity [65]. The overrepresented gene SEC61 Translocon Beta Subunit (SEC61B) is located only 60 kb downstream of TGFBR1. The primary function of the SEC61 complex is to translocate proteins across the endoplasmatic reticulum (ER) membrane, and, together with the Discoidin Domain Receptor 1, it regulates collagen IV synthesis, which is upregulated in fibrosis [66,67].

The gene *collagen type XV alpha 1 chain COL15A1* is likewise located within the same region under selection on ECA25 as *TGFBR1* and *SEC61B*, although not among the overrepresented genes. Mice deficient of *COL15A1* expression show exercise-induced skeletal myopathy and cardiovascular defects [68]. Recently, it was also shown that *COL15A1* gene expression is downregulated in skin fibroblasts derived from patients with the kyphoscoliotic form of Ehlers–Danlos syndrome (kEDS) [69]. This form of kEDS is caused by mutations in the genes *FK506–binding protein 14* (*FKBP14*) and/or *procollagen lysine, 2-oxoglutarate 5-dioxygenase 1* (*PLOD1*). Ehlers–Danlos syndrome has, in many cases, been associated with hypermobility, joint laxity and aortic aneurysms in humans [70]. The equine version of kEDS, i.e., Warmblood Fragile Foal Syndrome (WFFS), is caused by a recessive lethal missense mutation in *PLOD1,* where affected foals show severe skin fragility, and joint laxity [71]. Similarly, a region holding a gene with a known function in collagen regulation was found to be under selection in SJ horses by the XPEHH test, although not confirmed by *F_ST_*_._ This region (ECA12:11,851,991–11,854,261) harbors the gene encoding for the *zinc transporter solute carrier family 39 member 13* (*SLC39A13*). Mutations in this gene have been associated with collagen deficiency in the spondylocheiro dysplastic form of Ehlers–Danlos syndrome [72]. Overall, these findings suggest a strong selection on connective tissue functions in all sport horses, highlighting the importance of high mobility and relaxed locomotion patterns. This is, to some extent, in agreement with results from the previous study comparing SWB horses with a breed not selected for sport performance, in which other genes involved in muscle contraction and development were shown to be under putative selection in the SWB breed [16].

### 4.5. Selection for Mentality and Postsynaptic Signaling in SJ SWB Horses

In the SJ horses a total of 26 overrepresented genes were found in the GO term analysis for biological function and pathways. The genes were mainly related to organism development, gene regulation and cellular processes, and 20 of them were confirmed by *F_ST_* analysis. The largest region under putative selection in SJ horses is located on ECA22:27,935,550-31,789,708 with 24 ncRNAs and 37 protein-coding genes, of which nine are overrepresented protein-coding genes—*disc large homolog associated protein 4* (*DLGAP4*), *myosin light chain 9* (*MYL9*), *SAM and HD domain containing deoxynucleoside triphosphate triphosphohydrolase 1* (*SAMHD1*), *retinoblastoma 1* (*RBL1*), *SRC Proto–Oncogene/Non–Receptor Tyrosine Kinase* (*SRC*), *Rho GTPase activating protein 40* (*ARHGAP40*), *DNA Topoisomerase 1* (*TOP1*), *V-set and transmembrane domain containing 2 like* (*VSTM2L*)—and the uncharacterized protein *KIAA1755*. Some of the overrepresented genes, for example, *RBL1* and *SAMHD1*, have general functions in the cell, such as cell differentiation and nucleotide metabolism [73]. Most of the 24 ncRNAs are located in the 3’-end of this region and may be involved in the epigenetic regulation of gene expression of genes located within the region. It was previously shown that such epigenetic regulation could alter the gene expression of *DLGAP4* in cerebellar Purkinje cells, causing early-onset non-progressive cerebellar ataxia, and bipolar disorder in humans [74]. The membrane-associated guanylate kinase, encoded by the gene *DLGAP4*, the Rho GTPase activating protein 40, and the non-receptor tyrosine kinase encoded by the gene *SRC*, are all involved in the excitatory post-synaptic density (PSD), a large protein complex important in synaptic plasticity. Improper function of PSD has been associated with schizophrenia and bipolar disorder [75,76,77,78]. The proteins in PSD organize the post-synaptic cytoskeleton by linking glutamate receptors and signaling proteins to regulate receptor turnover in response to oscillating synaptic activity. This suggests that excitatory synaptic plasticity has an important function in SJ horses, possibly influencing the flexibility and reactive capacity needed when jumping. Genes involved in neurological control and signaling pathways were likewise found in shared ROH both among SWB horses and Hanoverian sport horses [14,16].

### 4.6. Selection for Growth and Muscle Function in SJ SWB Horses

The *SRC* gene has been shown to be involved in osteopetrosis due to non-functional osteoclasts [79]. The significant region on ECA8:10,891,925–11,400,093, comprises the genes *phosphatidylinositol transfer protein* (*PITPNB*) and *meningioma 1* (*MN1*), that overlap with a reported QTL for body growth trait in horses [80]. In humans, the two genes *PITPNB* and *MN1* are both involved in ossification processes during development [81]. The two genes, *TOP1* and *VSTM2L,* on ECA22, are both overrepresented in biological processes as well as cellular pathways, with a known function in skeletal–muscle development. In humans, mutations in *TOP1* have been associated with collagen buildup and growth of connective tissue [82], while *VSTM2L* was associated with carcass weight in Gir cattle, which implies a function in growth and muscle development [5].

The myosin light chain encoded by *MYL9* regulates muscle contraction by modulating ATPase activity [83]. The five genes *glycogen synthase kinase 3 beta* (*GSK3B*), *nuclear receptor subfamily 1 group I member 2* (*NR1I2*), *popeye domain containing 2* (*POPDC2*), *CD80 molecular* (*CD80*), and *Rho GTPase activating protein 31* (*ARHGAP31*), were overrepresented in SJ horses, and located within the same region on ECA19:41,459,599–41,959,599. The kinase encoded by *GSK3B* regulates several proteins involved in muscle glycogenolysis and glycolysis. One of these proteins is glycogen synthase (GS), encoded by the gene *glycogen synthase 1* (*GYS1*). A gain-of-function mutation in *GYS1* is responsible for the muscle disease polysaccharide storage myopathy (PSSM) in horses [84,85,86], and double-muscling in myostatin deficient cattle is a result of increased *GSK3B* phosphorylation of GS [85]. Furthermore, loss of *PTEN* function in mice also leads to the inactivation of *GSK3B* [87] that in turn, normally phosphorylates *ARHGAP31*. Dominant gain-of-function mutations in the *ARHGAP31* gene is associated to Adams–Oliver syndrome, where the patients show skin and limb defects [88]. The gene *Popeye domain containing 2* (*POPDC2*) encodes a membrane-bound protein, affecting potassium levels in striated muscles. Mice with null mutations in *POPDC2* develop muscle dystrophy and cardiac arrhythmia [89,90]. This ECA19 region, together with the regions on ECA8, and ECA22, appears to be under putative selection in SJ horses, where genes involved in muscle build-up and function are located.

### 4.7. Selection for CNS Reward System and Motoneuronal Control of Coordination in SJ SWB Horses

The *regulator of G protein signaling 17* (*RGS17*) gene is located within the significant region on ECA31:13,778,365–14,321,247 for SJ horses, and overlaps with a putative QTL for altitude adaptation in horses [91]. The *RGS17* modulates signaling of opioid receptors required in the endogenous reward system. The *RGS17* gene is located about 0.8 Mb downstream of the opioid receptor mu 1 (*OPRM1*) and may influence transcription of this gene. In humans, *RGS17* has been associated with risk for dependence on addictive substances [92]. This implies a possible function of *RGS17* in the equine endogenous reward system. In the genomic region in-between the *RGS17* and *OPRM1* genes, Doan et al (2012) and Kader et al (2016) reported one *indel* variation upstream of the gene *prothymosin alpha* (*PTMA*) and another *indel* covering a long non-coding RNA (lncRNA) [49,93]. Prothymosin alpha has been reported to influence locomotor activity, memory-learning, and anxiety behavior in mice, as well as modulating linker histone interaction to promote transcription [94,95]. It is therefore plausible that the genes *RGS17* and *OPRM1* are co-regulated and influenced by *PTMA* and lncRNA structural modifications.

Among the overrepresented biological processes, the gene *neuron navigator 2* (*NAV2*) and *developing brain homeobox 1* (*DBX1*) are located within the selected region on ECA7:89,669,652–91,203,503. Both genes are involved in locomotion patterns; *NAV2* was shown to be involved in motor coordination and balance in mice [96], while *DBX1* was shown to coordinate left–right locomotor activity [97]. In fact, *DBX1* is expressed in the same dI6 interneurons as *doublesex and mab-3 related transcription factors* (*DMRT3*) [98], known to affect locomotion pattern in horses [99]. Our findings indicate that showjumping horses are selected for neuromuscular control and coordination, and they might have a well-developed endogenous reward system that triggers them to seek fences to jump.

## 5. Conclusions

This study shows genetic divergence due to the specialization towards different disciplines in SWB horses. The two main subpopulations of SWB horses were, on average, moderately differentiated, except in eleven chromosomes where they showed more differentiation. Both measures of genomic diversity and extended haplotypes showed significant chromosomal regions with signatures of positive selection for either show jumping or non-show jumping performances. In show jumping horses, genes primarily related to the endogenous reward system, excitatory synaptic plasticity, neuromuscular control, and coordination, seemed to be under selection. On the other hand, genes involved in joint laxity, collagen build-up and muscle function are potentially under selection in non-showjumping horses, suggesting that there may be numerous genes involved in flexibility of movements in the performing horse. On top of this, many of the selected chromosomal regions comprise noncoding elements that are putative regulators of gene expression, implying a further dimension to the variability of the phenotypes selected in sport horses. Interestingly, in the present study, no fertility-related genes came up as potentially under selection, even if they were found in our previous study where we analyzed more general signatures of selection in the SWB population [16]. It seems reasonable to assume that fertility would be equally important when breeding sport horses for, e.g., show jumping and dressage. The results of this study point at many interesting genes to be validated by further functional genomic studies, such as, e.g., RNA-sequencing and/or detection of epigenetic DNA modifications. Genes that are differentially selected depending on discipline are most likely relevant for performance traits. In the future, this implies the possibility of a genomic approach in designing breeding schemes for specialized disciplines in sport horses.

## Figures and Tables

**Figure 1 genes-10-00976-f001:**
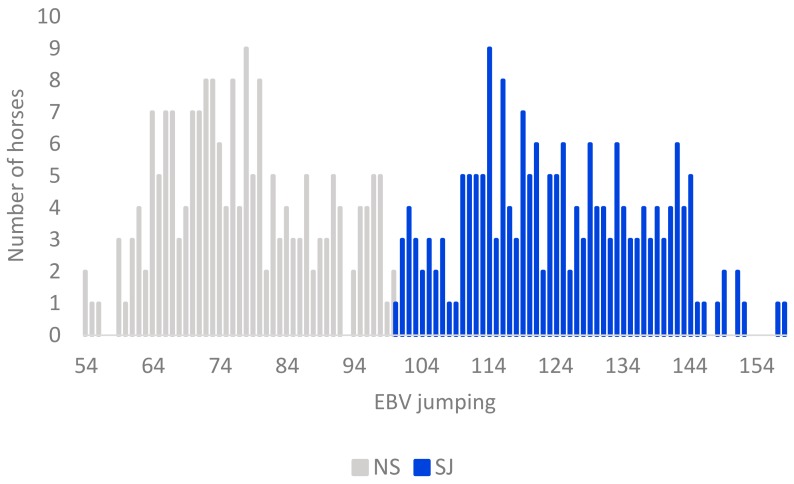
Distribution of estimated breeding values (EBV) for jumping performance in the 380 Swedish Warmblood horses included in this study. The distribution of EBVs for the horses assigned to the subpopulation non-show jumping horses (NS) are shown as grey bars, and, for show jumping horses (SJ), as blue bars.

**Figure 2 genes-10-00976-f002:**
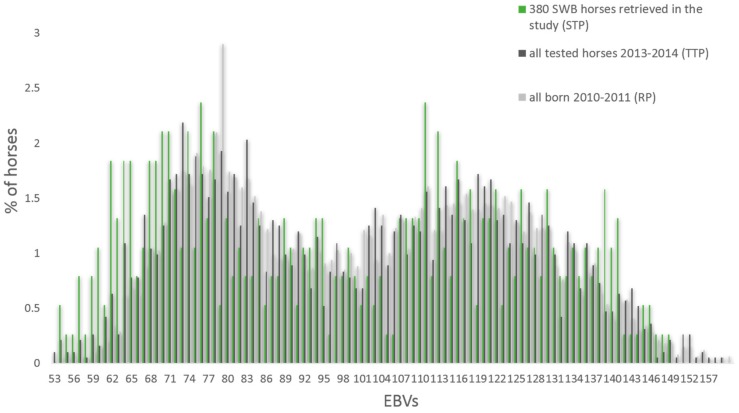
Percentage of horses with different estimated breeding values (EBVs) for show jumping in the studied horses (STP), all horses tested in young horse tests the same years (TTP), and all horses born in the same years’ cohort (RP).

**Figure 3 genes-10-00976-f003:**
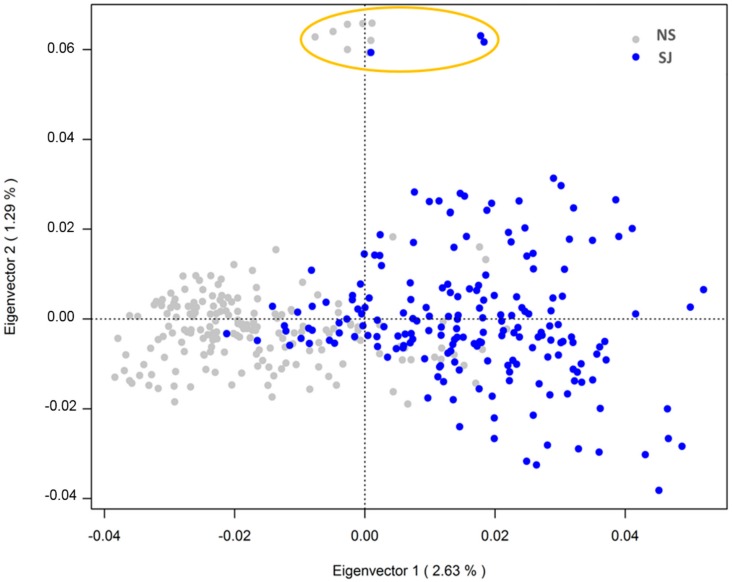
PCoA scatter plot of the first two Eigenvectors explaining genetic differentiation between non-show jumping (NS), which are gray colored, and show jumping (SJ) horses (blue colored) in the SWB population. The orange circle indicates the eleven outliers sired by the same stallion.

**Figure 4 genes-10-00976-f004:**
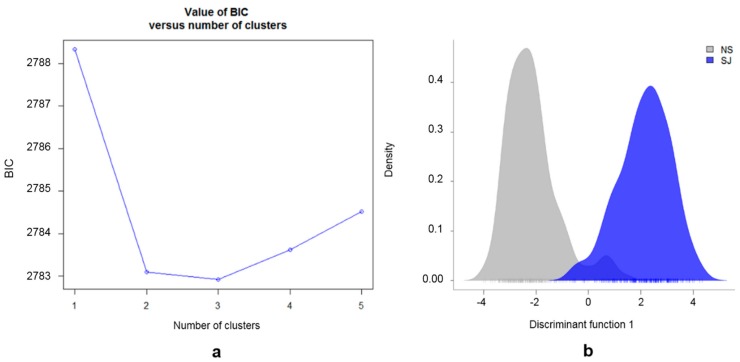
Genetic structure of SWB horses. (**a**) Inference of the number of clusters in SWB horses from analysis of genotype data based on K–means algorithm; (**b**) Density plot of individuals along the first discriminant function from the discriminant analysis of principal component (DAPC) for the two defined SWB subpopulations divided by EBVs for show jumping performance. The gray peak represents the non-show jumping horses (NS) and the blue peak represents the show jumping horses (SJ).

**Figure 5 genes-10-00976-f005:**
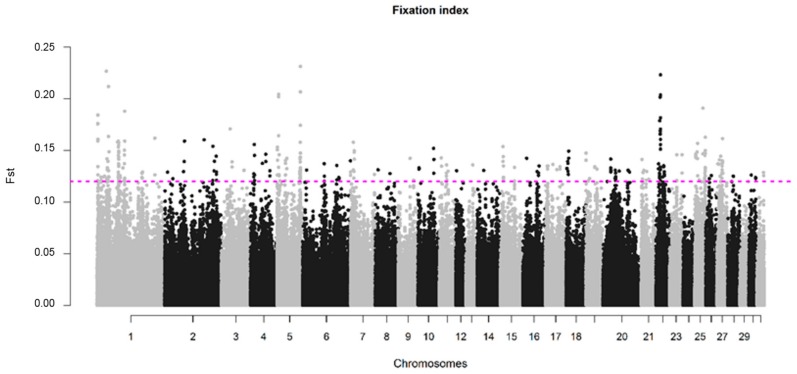
Genomic distribution of *F_ST_* values for SNPs plotted relative to their physical position within each autosomal chromosome. The cutoff to call a SNP as significant was defined as the highest 0.1% of the *F_ST_* values of the SNPs under analysis and is represented by the SNPs above the dotted pink line (*F_ST_* ≥ 0.12).

**Figure 6 genes-10-00976-f006:**
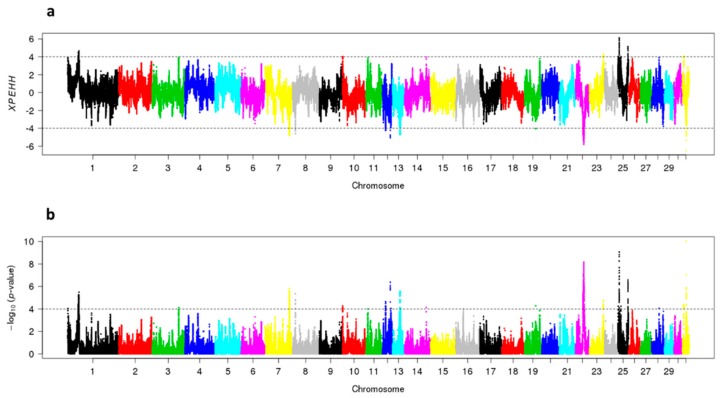
Cross population extended haplotype scores of the show jumping (SJ) and non-show jumping (NS) subpopulations: (**a**) XPEHH where positive values indicate positive selection in the NS subpopulation, and negative values indicate positive selection in the SJ subpopulation. An XPEHH value of +/−4 is considered significant and indicated by a dotted line; (**b**) log *p*-values of the XPEHH values.

**Table 1 genes-10-00976-t001:** Comparison between breeding values (EBVs), scored points at young horse evaluation, and proportion of horses competing, in the two assigned subpopulations show jumping (SJ) and non-show jumping (NS) horses.

Trait	SJ Horses	NS Horses
Mean EBV for show jumping performance	125 (13.4) *	77 (11.3) *
Mean EBV for dressage performance	94 (10.2) *	119 (21.0) *
Mean score for jumping technique	8.3 (1.2) *	5.6 (1.2) *
Mean score for gaits **	7.2 (0.6) *	7.3 (0.8) *
Mean score for the conformation trait “type”	7.9 (0.7) *	7.8 (0.7) *
Mean score for the conformation trait “head–neck–body”	7.7 (0.6) *	7.7 (0.7) *
Percentage of horses competing in show jumping at least at regional level	71%	11%
Percentage of horses competing in dressage at least at regional level	10%	38%

* Standard deviation within brackets, ** Mean score of walk, trot and canter.

**Table 2 genes-10-00976-t002:** Significant chromosomal regions found by the XPEHH test in showjumping horses (SJ). If the region overlapped with a significant *F_ST_*, the highest overlapping *F_ST_* value within the significant XPEHH region is shown.

Chromosome	Start Position (bp)	End Position (bp)	Length (kb)	Lowest Adjusted *p*–Value	Highest *F_ST_*
7	89,669,652	91,203,503	2918	0.0022	0.16
8	10,891,925	11,400,093	508	0.0040	0.13
12	11,601,991	12,104,261	502	0.0129	na
12	28,794,818	29,306,185	511	0.0009	0.13
13	27,280,931	27,830,947	550	0.0031	0.14
19	41,459,599	41,959,599	500	0.0247	0.13
22	27,935,550	31,789,708	3854	0.0008	0.14
31	13,778,365	14,321,247	543	0.00002	na

na = not applicable.

**Table 3 genes-10-00976-t003:** Overrepresented gene ontology (GO) term at *p*-value < 0.01 for the genes within regions indicated by both XPEHH and *F_ST_* to be under selection in SJ horses. GO terms are presented for both for biological processes and pathways.

PANTHER GOs	Genes	Fold Enrichment	Raw *p*–Value
**Slim Biological Process**			
Cell differentiation (GO:0030154)	*RBL1, DBX1, SRC, NR1I2*	5.68	<0.001
Multicellular organism development (GO:0007275)	*DBX1, DLGAP4, NAV2, MYL9, KIAA1755, VSTM2L, GSK3B, NR1I2, POPDC2, DPF2*	3.33	0.002
Small GTPase mediated signal transduction (GO:0007264)	*ARHGAP40, ARHGAP31, CDC42EP2*	6.25	0.004
Cellular response to lipopolysaccharide (GO:1901700.)	*NR1I2, CD80*	9.25	0.005
DNA conformation change (GO:0071103)	*CDCA5, TOP1*	19.6	0.005
Regulation of protein complex assembly (GO:0043254)	*ARHGAP40*	13.1	0.011
Purine nucleoside triphosphate metabolic process (GO:0009144)	*SAMHD1*	>100	0.014
Regulation of sister chromatid segregation (GO:0033045)	*CDCA5*	>100	0.014
Regulation of exit from mitosis (GO:0007096)	*CDCA5*	>100	0.014
Protein K48-linked ubiquitination (GO:0070936)	*SYVN1*	71.9	0.018
**Pathways**			
De novo pyrimidine deoxyribonucleotide biosynthesis (P02739)	*VSTM2L*	37.7	<0.001
CCKR signaling map (P06959)	*GSK3B, SRC*	6.78	<0.001
DNA replication (P00017)	*TOP1*	17.3	<0.001
Angiogenesis (P00005)	*GSK3B, SRC, NR1I2*	4.78	<0.001

**Table 4 genes-10-00976-t004:** Significant chromosomal regions found by the XPEHH test in non-showjumping horses (NS). If the region overlapped with a significant *F_ST_*, the highest overlapping *F_ST_* value within the significant XPEHH region is shown.

Chromosome	Start Position (bp)	End Position (bp)	Length (kb)	Lowest Adjusted *p*-Value	Highest *F_ST_*
1	40,592,555	43,510,660	2918	0.0230	0.21
23	51,321,303	51,844,470	523	0.0501	0.15
25	5,277,465	5,984,107	707	0.0002	0.15
25	37,351,887	37,886,254	534	0.0071	0.15

**Table 5 genes-10-00976-t005:** Overrepresented gene ontology term (GO) at *p*-value <0.01 for the genes within regions detected as potentially under selection in NS horses by the XPEHH and overlapped with *F_ST_* significant SNPs. GO terms are presented for both for biological processes and pathways.

PANTHER GOs	Genes	Fold Enrichment	Raw *p*-Value
**Slim Biological Process**			
SRP-dependent translational protein targeting to membrane (GO:0006614)	*SEC61B*	100.0	0.007
Posttranslational protein targeting to endoplasmic reticulum membrane (GO:0006620)	*SEC61B*	100.0	0.009
Intracellular protein transmembrane transport (GO:0065002)	*SEC61B*	88.7	0.012
Activin receptor signaling pathway (GO:0032924)	*TGFBR1*	81.3	0.013
Phospholipid biosynthetic process (GO:0008654)	*SGMS1*	75.0	0.014
**Pathways**			
Insulin/IGF pathway–protein kinase B signaling cascade (P00033)	*PTEN*	35.0	<0.001
p53 pathway feedback loops 2 (P04398)	*PTEN*	27.1	<0.001
Sulfate assimilation (P02778)	*PAPSS2*	100.0	<0.001
TGF-beta signaling pathway (P00052)	*TGFBR1*	14.5	<0.001

## Data Availability

The datasets in the current study was generated and analyzed in collaboration with the Swedish Warmblood Association and has a commercial value for them. The SWB horse data is therefore available from the corresponding author on reasonable request.

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
