# Peer review of "Genomic Divergence in Swedish Warmblood Horses Selected for Equestrian Disciplines"

_genes, 2019, doi:10.3390/genes10120976_

Round 1

Reviewer 1 Report

With regard to your manuscript «Genomic divergence in Swedish Warmblood horses selected for equestrian disciplines», my overall suggestion is to do a major revisión.

Comments and Suggestions for Authors.

First, and with regard to your manuscript «Genomic divergence in Swedish Warmblood horses selected for equestrian disciplines», my overall suggestion is to do a major revision. At a first glance, the manuscript does not have any important problem to be published, but considering one paper that appears in your bibliography, concretely [16] Ablondi et al.,«Signatures of selection in the genome of Swedish Warmblood horses selected for sport performance» of your research group, the samples and the markers described seem to be the same; moreover, the objectives seem to be very similar. The most important differences are with regard to the analytical methodologies and the obtained results. I consider your previous paper should be profusely cited not only in the Introduction but in all the manuscript along its different paragraphs. Obviously, the different chromosomes, affecting by signature selection, obtained in both papers should be cited, considered and analyzed.

Second, I will summarize the concrete suggestions should be considered:

  1. Line 57-62. There are two simple cites to your previous paper ([16]). As I stated above, it is not sufficient. There is an important backgroung in this published paper with regard this last manuscrip, and it should be widely developed.
  2. Line 89. I consider this should be the best place to explain if NS group can be considered as a dressage population or not. There is a paragraph in the Discussion that explain it, but it is not clear along the paper.
  3. Line 105. How do you quantify and qualify the obtained DNAs? Spectrophotometry, fluorometry,...?
  4. Line 110. If there is a reason for excluding markers from the sexual chromosome, I consider that should be, at least, briefly cited.
  5. Line 177-179. The values cited from the Table 1 should be also introduced (in brackets) in the text. This is the best way to understand what you want to say at the same time the reader can observe them in the table.
  6. Line 203. It is not correct to say that BIC analyses indicated two main subpopulations. It seems to be three subpopulations (SJ, NS and, probably, the 23 individuals in the overlapping area). Moreover, in the Discussion you comment this fact, so why not to introduce it here?.
  7. After Figure 5 and before line 228. It seems to be clear there several animals that do not fit the breed (or population) criterium: the 23 individuals with a high level of Thoroughbred blood, and those do not cluster with their expected subpopulation (Figure 5). These animals have been included in the following analyses or not? A brief explanation should be done at this point.
  8. Line 228 (paragraph 3.3). There is no cite regarding the Results from the previous paper. There is no comment concerning ECA1 (chromosome under signature selection in both papers) nor about ECA4 and ECA6 (from the previous paper only).
  9. Line 293 (4.1 Horses selected for this study). The first paragraph contains some data that could fit better in the Introduction. This would help readers to better undestand if NS subpopulation is related with animals for dressage or not.
  10. Line 348. I am agree when you state «Both FST and XPEHH are effective when the traits selected in one subpopulation are not selected in the other subpopulation, resembling the case of show jumping traits» I wonder what will occur if you used the data for the Exmoore pony as you did it before. It would be interesting and more comparable with your previous paper.
  11. Paragraphs 4.4 and 4.5 from Discussion part. The genes SEC61B, SAMHD1, RBL1 and uncharacterized protein KIAA1755, have appeared as overrepresented, but there is no further explanation about them.
  12. Line 497. When you state «further functional genomic studies», what do you mean? Epigenomics, RNAseq, both?
  13. Discussion and Conclusion. As stated above, the discussion of the results and, of course, the derived conclusions should include the results from the previous paper.

Author Response

First, and with regard to your manuscript «Genomic divergence in Swedish Warmblood horses selected for equestrian disciplines», my overall suggestion is to do a major revision. At a first glance, the manuscript does not have any important problem to be published, but considering one paper that appears in your bibliography, concretely [16] Ablondi et al.,«Signatures of selection in the genome of Swedish warmblood horses selected for sport performance» of your research group, the samples and the markers described seem to be the same; moreover, the objectives seem to be very similar. The most important differences are with regard to the analytical methodologies and the obtained results. I consider your previous paper should be profusely cited not only in the Introduction but in all the manuscript along its different paragraphs. Obviously, the different chromosomes, affecting by signature selection, obtained in both papers should be cited, considered and analyzed.

Answer:

Dear Referee, thank you very much for your feedback. We agree with you that, in the current work, we should refer more to our previous paper. The Ablondi et al. 2019 BMC Genomics paper was accepted just a few days prior to the deadline for the special issue “Equine Genetics”, thus, due to limited time, we were unable to include more about it in the previous version of the manuscript.

Thanks to your feedback we have now integrated and cited our previous work along the manuscript (please find details on Line numbers in the following answer). Nevertheless, even if in both manuscripts we searched for signatures of selection, as you mentioned, the applied methodologies were different. Likewise, the objectives – and consequently the expected results -are different yet complementary.

In the previous paper we searched for signatures of selection for general athletic sport performance in SWB horses (regardless of specific discipline) in comparison to a breed not selected for sport performance. In the present manuscript we instead aimed to investigate how the selection pressure has shaped the genome of SWB horses into two subpopulations diverging by discipline. Both disciplines require athletic performance, but of different types. Our present analyses resulted in 1) a deep characterization of the current genetic divergence in the SWB breed, 2) the discover of the genomic regions under potential selection in two different sport disciplines.

In other words, in the previous study (as well as in few more already published studies performed in other horse breeds) we addressed differences between breeds rather than subpopulations formed by selection within breed. We consider the SWB to be especially well suited for this latter type of study since all tested SWB horses are assessed and evaluated for both show jumping and dressage traits even if they are bred for different disciplines.

Lastly, we do not expect to find signatures of selection in the same genomic regions when comparing between breeds as when comparing subpopulations within breed. We hypothesized that the regions we found to be selected in our previous paper, are important for sport horses, regardless of discipline.

Line 57-62. There are two simple cites to your previous paper ([16]). As I stated above, it is not sufficient. There is an important backgroung in this published paper with regard this last manuscrip, and it should be widely developed.

Answer:

Yes, we agree, and we further included our previous manuscript in L71 to L76, L359 to L364, L432 to L434, L458 to L460, and from L523 to L527.

Line 89. I consider this should be the best place to explain if NS group can be considered as a dressage population or not. There is a paragraph in the Discussion that explain it, but it is not clear along the paper.

Answer:

We have tried to address your concern at L97 to L98. However, we hesitate to call the NS group directly as “dressage horses” since we formally divided the two groups based on exclusively the EBV for show-jumping. It is true that horses with a low EBV for show-jumping are mainly dressage horses but we cannot state it with 100% certainty for all of them.

Line 105. How do you quantify and qualify the obtained DNAs? Spectrophotometry, fluorometry,...?

Answer:

The DNA concentrations of the sample were measured by fluorescence using the Quant-iTTM PicoGreenTMdsDNA Assay Kit. This information is now added to the manuscript at L113 to L116.

Line 110. If there is a reason for excluding markers from the sexual chromosome, I consider that should be, at least, briefly cited.

Answer:

We added the following sentence at L121 to L123:

“Allosomes were not considered since no Y chromosome data were available and thus allosomes would not enable homozygosity-based analyses in male individuals”

Line 177-179. The values cited from the Table 1 should be also introduced (in brackets) in the text. This is the best way to understand what you want to say at the same time the reader can observe them in the table.

Answer:

Yes we agree, you can now find the values in brackets. 

Line 203. It is not correct to say that BIC analyses indicated two main subpopulations. It seems to be three subpopulations (SJ, NS and, probably, the 23 individuals in the overlapping area). Moreover, in the Discussion you comment this fact, so why not to introduce it here?

Answer:

We agree that the BIC value for N.= 3 is slightly lower (thus better) than the BIC for N.= 2. However, when calculating the ∆BIC we do not have a significant difference (It is generally called significant when the delta BIC is higher than 2). Thus, we preferred to stick to show only significant results in the “result section” and in the “discussion” broaden the topic to a more general interpretation of the results.

After Figure 5 and before line 228. It seems to be clear there several animals that do not fit the breed (or population) criterium: the 23 individuals with a high level of Thoroughbred blood, and those do not cluster with their expected subpopulation (Figure 5). These animals have been included in the following analyses or not? A brief explanation should be done at this point.

Answers:

Yes, those animals were included in the following analyses as we believe they likewise represent the current SWB population, where there is still a certain degree of influence from thoroughbred blood lines as well as horses from other warmblood studbooks.

Line 228 (paragraph 3.3). There is no cite regarding the Results from the previous paper. There is no comment concerning ECA1 (chromosome under signature selection in both papers) nor about ECA4 and ECA6 (from the previous paper only).

Answer:

In short, there are no genomic regions that do exactly overlap between the previous paper (Ablondi et al 2019, BMC Genomics), and our current study. We have now included a more thorough discussion on the comparison between the results found in the two studies at L359 to L364, L432 to L434, L458 to L460, and from L523 to L527.

Line 293 (4.1 Horses selected for this study). The first paragraph contains some data that could fit better in the Introduction. This would help readers to better undestand if NS subpopulation is related with animals for dressage or not.

Answer:

We have now moved some of this description to the introduction (L44 to L49).

Line 348. I am agree when you state «Both FST and XPEHH are effective when the traits selected in one subpopulation are not selected in the other subpopulation, resembling the case of show jumping traits» I wonder what will occur if you used the data for the Exmoore pony as you did it before. It would be interesting and more comparable with your previous paper.

Answer:

Yes, we agree with you that the comparison of SJ and NS horses with Exmoor pony would be interesting. However, we think that adding these analyses would not benefit the readability of our work. Also, in the current study we did not focus on general genomic differences between horses selected for sport performance or not, but instead on differences between horses selected for different disciplines within a breed of athletic sport horses. Since no genomic regions were overlapping between the previous and current study, we believe that the suggested extra analysis would not gain any extra information.

Paragraphs 4.4 and 4.5 from Discussion part. The genes SEC61B, SAMHD1, RBL1 and uncharacterized protein KIAA1755, have appeared as overrepresented, but there is no further explanation about them.

Answer:

To reduce the complexity of the discussion, we tried to condense the number of genes to discuss, and therefore omitted some genes with more general cell functions. We now added some discussion about the gene SEC61B that is overrepresented in NS horses, and involved in regulation of collagen IV synthesis. We also added some information about the GTPase activator ARHGAP40 in the discussion, and mention the general functions of RBL1, and SAMHD1.

Line 497. When you state «further functional genomic studies», what do you mean? Epigenomics, RNAseq, both?

Answer:

We now added a more detailed discussion on L527 to L531.

Discussion and Conclusion. As stated above, the discussion of the results and, of course, the derived conclusions should include the results from the previous paper.

Answer:

We have broadened our conclusion including a comparison with the previous study (L523 to L527). As far as the discussion is concerned, please find the added line numbers in the previous answers.

Reviewer 2 Report

The authors of the manuscript “Genomic divergence in Swedish Warmblood horses selected for equestrian disciplines” described the identification of genomic regions influenced by selective pression in Swedish Warmblood horses between animals with high and low breeding values for jumping performance. The results reported here are relevant and may help to better understand the population structure and the biological processes regarding the jumping performance in Swedish Warmblood horses. However, several point must be better explained and clarified across the manuscript. Please, check my comments bellow:

Major comments:

Introduction

In general, the introduction is well written. However, before the last paragraph it would be helpful the addition of some sentences explaining the main question and/or the missing points in the literature regarding the selective pressure in jumping horses that the work aimed to address.

Material and methods:

The authors must provide the detailed procedure to estimate the breeding values. What were the models? Which software was used? Was used genomic information? The reference used in this point is about the guidelines for the genomic evaluation, not about the EBV calculation for the animals used here.

In lines 103-104 the authors described the procedure used for DNA extraction. However, the authors informed that “The samples were incubated at 56°C, 1500 rpm for 2 hours”. It is not clear if the samples were incubated for 2 hours in this temperature and subsequently centrifuged in 1500 rpm. Usually, in DNA extraction protocols samples are not centrifuged for so long time. Please, review this sentence.

In lines 118-119, please provide the genomic relationship metric used to calculate the genomic distances. I assume that Kinship was used. However, this must be clearly showed.

In lines 134-135 the authors introduced the methods used to study the selective pressure between the group of animals. However, it well established that the relationship and/or family structure among animals in a sample can affect the Fst estimates. What is the relationship level among the animals within each subpopulation? What are the limitations of Fst using related animals? The authors must consider the impact of cryptic relatedness in genomic estimates. I would suggest the authors to evaluate the level of relatedness among the animals in the dataset and if the relatedness level is higher than expected by chance (i.e., third degree) to apply a resampling strategy to reduce this relationship. Consequently, the results can be compared and evaluate the impact of the relationship over the genomic estimates.

In lines 143-144 the authors must clearly show which criteria was used to filter the phased haplotypes.

Results

In lines 196-197 the authors should clearly highlight these outliers in the plot.

In lines 263-268, the authors provide the results about the enrichment analysis. These results are one of my main concerns about the manuscript. Regarding the p-values used for the enrichment analysis, was performed any multi-testing correction? If yes, this must be showed. If not, the authors must inform it. The absence of significant GO or pathways is not a problem. The authors want to identify possible candidate genes in the regions of signature selection. Therefore, it is possible that the most interesting terms will not be enriched. This is very common to be observed in genome-wide association studies. The enrichment for biological processes or pathways is expected in studies such as RNA-sequencing, once the co-expressed genes show a trend to be acting in similar biological functions in a much more evident regulatory network. However, in genomic studies, it is possible that the results will highlight the main effects, not all the (or the majority) of the regulatory network. Therefore, it is possible that candidate genes will be allocated in GO or pathways that are not enriched. Additionally, in the tables with the enrichment results, mainly very broad terms are presents. This reinforce the topic discussed before, once, broad terms tend to group more genes. Another interesting point to be highlighted is the presence of several terms enriched with only one related gene. What this mean? Is this a real enrichment? The enrichment concept is based on the higher probability to identify among the list of candidate genes a number of gene related with a specific term then expected by chance. Therefore, in these cases, few genes are associated with these biological processes. What this means? Did the authors use the horse genome as the reference for the enrichment analysis? If yes, probably the enrichment results are highly influenced by the quality of the GO and pathway annotation. The use of tools such as Blast2GO or the use of better annotated genomes from evolutionary close species might help to avoid the issued regarding the annotation quality.

Discussion:

In lines 319-320 the authors discuss the interpretation of the clustering based on the PCA analysis. It is important to highlight that not only the specialization processes might be influencing the PCA results. Therefore, use this result as a decision point to a possible complete specialization process, it is not recommended. Additionally, as showed in lines 191-195, the percentage of the variance explained by the two main components are very low. The use of resampling strategies, as suggested before in this review might help to increase this percentage through the reduction of cryptic relatedness and confusing factor. However, this must be weighted based on the sample size reduction as well.

In lines 349-350 the authors discussed the possible functional impacts of the variants with high Fst. However, it is well-known that usually the markers present in genotyping arrays are not selected based on functional impact. Usually, these markers are selected based on the good representation of the linkage disequilibrium structure of the region which these markers are mapped. Therefore, unless there are any clear information about the function of these variants, to discuss the function of these alleles as the reason for the high Fst is not indicated using genotyping chips. In general, the causal variants are in LD with the markers in the chip. In other words, the high Fst observed for these SNPs are an indirect effect of the LD between the marker in the genotyping chip and the real causal variants. This is one of the reasons why the Fst and other genomic metric may vary substantially between populations for the same marker even when similar traits are evaluated.

Another recurrent point across the discussion is the use of the term “enriched genes”. The genes are not enriched in the analysis. The processes which these genes are associated were identified as enriched. Tis must be fixed across the discussion.

Minor comments:

Lines 65-68: This kind of sentence can be removed from the introduction and maintained only in the material and methods section.

Figure 3: The legend must describe the colors. Moreover, the plot legend must be edited. The numbers “1” and “2” are confusing. It is possible to remove the title and to add “NS horses” and “SJ horses” in front of the bullet points.

Figure 4: The title and ticks of the plot are very little.

Figure 5: This figure is not very descriptive. I would suggest remove it from the manuscript and to inform the probabilities in a supplementary table.

Line 233: Is “outlier” the better word to describe the SNPs with higher Fst?

Lines 252-253: Which QTLs are overlapping with the candidate regions?

Table 2: The title of this table is not clear. The authors should review it in order to provide a clear explanation about the table content.

Lines 298-299: Reference.

Line 342: The authors should standardize the point of discussion in this sentence. If the sentence has an introduction about the number of genes in complex traits, the conclusion also must be about the number of genes, not the number of variants.

Line 360: Selection signature for what?

Lines 379-381: Use parenthesis to clearly separate the acronym of the genes.

Line 388: Is the PAPSS2 gene or variants mapped in this gene the responsible for different odds ratio?

Lines 391-392: This doesn’t mean anything. Which biological processes?

Lines 465-466: What are the evidences for this? Only proximity?

Author Response

The authors of the manuscript “Genomic divergence in Swedish Warmblood horses selected for equestrian disciplines” described the identification of genomic regions influenced by selective pression in Swedish Warmblood horses between animals with high and low breeding values for jumping performance. The results reported here are relevant and may help to better understand the population structure and the biological processes regarding the jumping performance in Swedish Warmblood horses.

Answer:

Dear referee, thank you very much for your positive overall evaluation and thank you for your useful comments. Please find below our answers.

Introduction

In general, the introduction is well written. However, before the last paragraph it would be helpful the addition of some sentences explaining the main question and/or the missing points in the literature regarding the selective pressure in jumping horses that the work aimed to address.

Answer:

Yes, we agree and we added few more sentences about the novelty of the presented study at L71 to L76.

Material and methods:

The authors must provide the detailed procedure to estimate the breeding values. What were the models? Which software was used? Was used genomic information? The reference used in this point is about the guidelines for the genomic evaluation, not about the EBV calculation for the animals used here.

Answer:

Sorry, we wrongly assigned the reference to the paper where all the requested details are presented. We have now corrected the reference information in the manuscript (reference number 4: Viklund, Å., Näsholm, A., Strandberg, E. and Philipsson, J. 2011. Genetic trends for performance of Swedish Warmblood horses. Livest. Sci, 144, 113-122.)

In lines 103-104 the authors described the procedure used for DNA extraction. However, the authors informed that “The samples were incubated at 56°C, 1500 rpm for 2 hours”. It is not clear if the samples were incubated for 2 hours in this temperature and subsequently centrifuged in 1500 rpm. Usually, in DNA extraction protocols samples are not centrifuged for so long time. Please, review this sentence.

Answer:

Sorry for the confusion, the samples were vortexed, not centrifuged.

We changed to: “The samples were vortexed at 1500 rpm for 2 hours in 56°C, followed by heat inactivation of Proteinase K at 96°C for 10 minutes.”

In lines 118-119, please provide the genomic relationship metric used to calculate the genomic distances. I assume that Kinship was used. However, this must be clearly showed.

Answer:

For the PCoA a dissimilarity matrix is necessary which expresses genomic distance. In other words, we used a matrix of IBS counts as a distance matrix. The distance between two samples is returned as the proportion of allele comparisons which are not IBS.

In lines 134-135 the authors introduced the methods used to study the selective pressure between the group of animals. However, it well established that the relationship and/or family structure among animals in a sample can affect the Fst estimates. What is the relationship level among the animals within each subpopulation? What are the limitations of Fst using related animals? The authors must consider the impact of cryptic relatedness in genomic estimates. I would suggest the authors to evaluate the level of relatedness among the animals in the dataset and if the relatedness level is higher than expected by chance (i.e., third degree) to apply a resampling strategy to reduce this relationship. Consequently, the results can be compared and evaluate the impact of the relationship over the genomic estimates.

Answer:

As mentioned from L87 to L90, the horses in this study descended from 145 sires with 1-11 offspring each, and 372 mares with 1-2 offspring each. In order to accomplish your comment on relationship level, we calculated the average relationship within and between groups (group 1=NS, group 2 =SJ) based on pedigree data: NS-NS = 0.02525; SJ-NS = 0.0095; and SJ-SJ = 0.0357. The average relationships were as expected, higher within than between subgroups (in agreement with what was shown by the genomic analyses as well) but not that high overall. This is because the SWB is a breed with a large founding population, a semi-open studbook, and it has not experienced any recent bottlenecks. As a result, the average relationship within the breed is quite low compared to many other breeds. In addition, all horses are parentage tested and the pedigrees are deep with a high pedigree completeness. Therefore, we believe that the problems with cryptic relatedness should not be an issue in our data set. Considering that our data is already of limited size, we would lose a lot of power if we remove animals and decrease the sample size even further. Lastly, the EBV distribution in our sample compared to the population as a whole, clearly show that we have a representative sample of the population. For those reasons, we did not apply any resampling in this study.

In lines 143-144 the authors must clearly show which criteria was used to filter the phased haplotypes

Answer:

Agree, we have now added the description of the filter used to check the phased haplotype in the manuscript (L 156 to L 157).

In lines 196-197 the authors should clearly highlight these outliers in the plot.

Answer:

We decided to add a circle in the figure to better highlight the outliers. This sire has over the years produced top performing offspring in all equestrian disciplines.

Results

In lines 263-268, the authors provide the results about the enrichment analysis. These results are one of my main concerns about the manuscript. Regarding the p-values used for the enrichment analysis, was performed any multi-testing correction? If yes, this must be showed. If not, the authors must inform it. The absence of significant GO or pathways is not a problem. The authors want to identify possible candidate genes in the regions of signature selection. Therefore, it is possible that the most interesting terms will not be enriched. This is very common to be observed in genome-wide association studies. The enrichment for biological processes or pathways is expected in studies such as RNA-sequencing, once the co-expressed genes show a trend to be acting in similar biological functions in a much more evident regulatory network. However, in genomic studies, it is possible that the results will highlight the main effects, not all the (or the majority) of the regulatory network. Therefore, it is possible that candidate genes will be allocated in GO or pathways that are not enriched. Additionally, in the tables with the enrichment results, mainly very broad terms are presents. This reinforce the topic discussed before, once, broad terms tend to group more genes. Another interesting point to be highlighted is the presence of several terms enriched with only one related gene. What this mean? Is this a real enrichment? The enrichment concept is based on the higher probability to identify among the list of candidate genes a number of gene related with a specific term then expected by chance. Therefore, in these cases, few genes are associated with these biological processes. What this means? Did the authors use the horse genome as the reference for the enrichment analysis? If yes, probably the enrichment results are highly influenced by the quality of the GO and pathway annotation. The use of tools such as Blast2GO or the use of better annotated genomes from evolutionary close species might help to avoid the issued regarding the annotation quality.

Answer:

“the p-values used for the enrichment analysis, was performed any multi-testing correction?”

We agreed that a better explanation of the p-values presented in the GO term analyses is needed. As far as we know, it is very difficult to find a consensus on how to perform Panther analysis and it is still an ongoing trivial debate. Generally, as far as we understood, due to their structure, the GO terms, are not well suited for a multiple testing correction. (GO terms are highly dependent - terms on the same branch + one gene is annotated in several terms). Thus, we decided not to implement any multiple testing corrections. We clarified “Raw p-value” in the manuscript.

“it is possible that the most interesting terms will not be enriched.”

Yes, we are aware of this risk, and we do discuss some genes that are not significantly enriched by GO analysis, but have other support from literature or QTLdb etc. We did try to reduce the length of the discussion, and avoid false positives by being quite strict in what genes we did actually discuss about. We are aware of the limitations of GO terms but still, it is widely used in similar studies. We are absolutely aware of that the GO terms do not highlight regulatory networks in the same way as annotated genes. We believe that the variation seen in many traits involved in performance of sport horses are actually a result of regulatory genetic variation, why this needs to be further studied in the regions we have found to be under putative selection in SJ and NS horses.

“broad terms tend to group more genes,“

Yes, this is true, but we have also considered what genes are located in each region, and if there are several genes in enriched GO terms within the same region, we believe this is an even stronger evidence of selection in this area. Genes in non-enriched GO terms could be co-selected and may as well contribute to the phenotype. Further studies are needed to elucidate what genes are the most important contributors in each putative region.

“several terms enriched with only one related gene” What this mean? Is this a real enrichment?”

A specific GO term could have one single gene assigned. This means that there are most probably no other genes in our study that do have the same specific gene ontology, but such genes could as well be included in several GO terms. The genes that are not annotated cannot be explain further by this analysis, but could be starting points for other studies.

We are aware of the pitfall of the GO term and indeed we used the results from the GO term as a guidance to find possible patterns of positive selection, and we completed our discussion with genes already found in the literature e.g. QTLs etc.

Further studies are needed also in this case.

“horse genome as the reference for the enrichment analysis?”

Yes, the horse genome was used as reference for the overrepresentation test (we added this detail in the text). We agree that the results might be influenced by the yet scarce annotation compared to human or mouse, however so far, the equine annotation has been broadly used in previous studies. Although we think it could be very interesting to compare GO term results using better annotated genomes from evolutionary close species, but to avoid a very long manuscript we prefer to keep the analyses as they are. Especially because, as previously mentioned, the GO term results were merely used as starting point for our discussion, and we complement our discussion with genes already found in the literature e.g. QTLs etc.

In lines 319-320 the authors discuss the interpretation of the clustering based on the PCA analysis. It is important to highlight that not only the specialization processes might be influencing the PCA results. Therefore, use this result as a decision point to a possible complete specialization process, it is not recommended. Additionally, as showed in lines 191-195, the percentage of the variance explained by the two main components are very low. The use of resampling strategies, as suggested before in this review might help to increase this percentage through the reduction of cryptic relatedness and confusing factor. However, this must be weighted based on the sample size reduction as well.

Answer:

We agree that the subpopulations are not completely separated, and we have added a sentence to explain this more clearly (L329 to L331).

Please see our response to your previous comment about cryptic relatedness and average relationship.

In lines 349-350 the authors discussed the possible functional impacts of the variants with high Fst. However, it is well-known that usually the markers present in genotyping arrays are not selected based on functional impact. Usually, these markers are selected based on the good representation of the linkage disequilibrium structure of the region which these markers are mapped. Therefore, unless there are any clear information about the function of these variants, to discuss the function of these alleles as the reason for the high Fst is not indicated using genotyping chips. In general, the causal variants are in LD with the markers in the chip. In other words, the high Fst observed for these SNPs are an indirect effect of the LD between the marker in the genotyping chip and the real causal variants. This is one of the reasons why the Fst and other genomic metric may vary substantially between populations for the same marker even when similar traits are evaluated.

Answer:

To get Fst values that were more comparable to our XPEHH results, we analyzed the Fst divergence by sliding windows, instead of by single SNP Fst.

We are absolutely aware of that the majority of the SNPs in the chip are neutral mutations without any functional roles but the genes in or close to where they are located in the detected genomic regions, might have a function. The markers are selected to be neutral variants in as many breeds as possible, regardless of usage of the breed or function of the marker. The markers are usually noncoding mutations, and not causative mutations for any trait. The reviewer is correct in that the markers are in LD with causative mutations. Since the markers are selected to be polymorphic, a region of lower heterozygosity, is believed to be a result of selection on that region. The phenotypic effect is not caused by the markers themselves, but by causative mutations in LD with the markers. To identify the causative mutations in all genes under putative selection will generate fuel to a large amount of further studies.

As the SNP markers are included in the SNP-chip with the criteria to be highly polymorphic, a lower level of heterozygosity, even strengthens the probability that there is in fact a selection pressure on the corresponding genomic regions.

Another recurrent point across the discussion is the use of the term “enriched genes”. The genes are not enriched in the analysis. The processes which these genes are associated were identified as enriched. Tis must be fixed across the discussion.

Answer:

Thanks, we corrected this point throughout the manuscript!

Lines 65-68: This kind of sentence can be removed from the introduction and maintained only in the material and methods section.

Answer:

Yes, we agree and we remove it from the introduction.

Figure 3: The legend must describe the colors. Moreover, the plot legend must be edited. The numbers “1” and “2” are confusing. It is possible to remove the title and to add “NS horses” and “SJ horses” in front of the bullet points.

Answer:

Yes, thank you for the suggestions. We modified the figure according to your feedback.

Figure 4: The title and ticks of the plot are very little.

Answer:

Yes agree, we made them bigger. Thank you for your remark.

Figure 5: This figure is not very descriptive. I would suggest remove it from the manuscript and to inform the probabilities in a supplementary table.

Answer:

Yes, we agree and we have now changed this figure into a supplementary file.

Line 233: Is “outlier” the better word to describe the SNPs with higher Fst?

Answer:

We agree and we changed “outliers” to “significant”.

Lines 252-253: Which QTLs are overlapping with the candidate regions?

Answer:

We added the main functions of the QTLs in the result section but we kept a further discussion of them in the discussion section.

Table 2: The title of this table is not clear. The authors should review it in order to provide a clear explanation about the table content.

Answer:

We have now added information in the table head.

Lines 298-299: Reference.

Answer:

We have now added the reference also in the first sentence that it refers to.

Line 342: The authors should standardize the point of discussion in this sentence. If the sentence has an introduction about the number of genes in complex traits, the conclusion also must be about the number of genes, not the number of variants.

Answer:

We clarified in the text that time is needed to reach fixation, especially for complex traits where the desired phenotype results from mutations in a network of genes, rather than from monogenic point mutations (L353 to L354).

Line 360: Selection signature for what?

Answer:

Added at line L375 to L376.

Lines 379-381: Use parenthesis to clearly separate the acronym of the genes.

Answer:

Corrected

Line 388: Is the PAPSS2 gene or variants mapped in this gene the responsible for different odds ratio?

Answer:

Yes, in humans there is an OD of 1.37 to be an exerciser for the T-allele of SNP rs10887741. We did not add this information into the discussion as we thought it was in to much detail.

Lines 391-392: This doesn’t mean anything. Which biological processes?

Answer:

We added the biological and pathways which those genes are involved in.

Lines 465-466: What are the evidences for this? Only proximity?

Answer:

We agree that a less strong conclusion must be made at this point. We corrected the text accordingly.

Reviewer 3 Report

In abstract: suggest inserting closing sentence: "Study shows genetic divergence due to the specialization towards different disciplines in SWB horses...". 

In 2. References - recommend a new look on the "Retrived in date " 

In 6. References - talks about Quarter Horses and is cited as selection of cattle. 47 "Recent examples of genomic studies have shown the benefits of studying genetic subpopulations."- add that it is a Horses briding

86 references 4. from 2018 in cited as "Breeding values from the latest routine genetic evaluation (2018)..." or is it just "mullti-trait animal model and based" from Viklund et al. (2008)

92 what year is it from 1,540 horses tested? From Figure 2 all tested horses 2013-2014 (TTP) 

What is the impact of the gender of the horse on the sporting result "80 The horses (182 males and 198 females)" ? 

183 "Until October 2018, 155 of the horses ..." is there any data on hew successful the horses are in show jumping and dressage? And the horses (? males and ? females) from 155 show jumping 91 dressage and 18 had competed in both disciplines? 

Author Response

Reviewer 3
In abstract: suggest inserting closing sentence: "Study shows genetic divergence due to the
specialization towards different disciplines in SWB horses...".
Answer:
Yes, we agree and we added the suggested closing sentence at L23 to 25 of the abstract.
In 2. References - recommend a new look on the "Retrived in date "
Answer:
Agree and rechecked in data 28/10/2019
In 6. References - talks about Quarter Horses and is cited as selection of cattle. 47 "Recent
examples of genomic studies have shown the benefits of studying genetic subpopulations."-
add that it is a Horses briding
Answer:
Thanks, corrected
86 references 4. from 2018 in cited as "Breeding values from the latest routine genetic
evaluation (2018)..." or is it just "mullti-trait animal model and based" from Viklund et al.
(2008)
Answer:
Yes sorry, we corrected the location of the reference.
92 what year is it from 1,540 horses tested? From Figure 2 all tested horses 2013-2014 (TTP)
Answer:
Thank you for the good remark! We added this detail also in the text.
What is the impact of the gender of the horse on the sporting result "80 The horses (182
males and 198 females)" ?
Answer:
We would like to mention that a correction for the sex effect is used in the EBVs estimation.
When we looked at later performance, we saw that the sexes were well represented. Please
have a look at our following answer for more details.
183 "Until October 2018, 155 of the horses ..." is there any data on hew successful the
horses are in show jumping and dressage? And the horses (? males and ? females) from 155
show jumping 91 dressage and 18 had competed in both disciplines?
Answer:
Both for dressage and show jumping horses the gender distribution was 47% mares and 53%
geldings and stallions (mainly geldings). If we only look at the 18 horses competing in both
disciplines, there were 8 mares (44%) and 10 males (56%).
How successful were the horses?
Well, out of the 155 horses that had competed in show jumping 89 horses had, as best result,
been placed at easy level, 44 horses had been placed at intermediate level and 3 had been
placed at advanced level. For the 91 horses competing in dressage the numbers were 35
placed at easy level and 16 at intermediate level (no horse placed at advanced level).
Nevertheless, those horses are still young, so they have not yet reached higher levels in
competition, and dressage horses tend to reach the higher levels at a higher age than show
jumping horses does.

Reviewer 4 Report

As a reviewer I like the manuscript. Overall bit elaborated. Introduction is well written, however if this study performed first time is Swiss Warmblood horses then you can mention that. Also in conclusion can you mention any candidate gene or non-coding region want to study further? In reference section 49 and 90 are same reference. Also in 90 the title of the paper written twice. 

Author Response

As a reviewer I like the manuscript. Overall bit elaborated.

Answer:

Dear referee, thank you for your positive evaluation of our work.

Introduction is well written, however if this study performed first time is Swiss Warmblood horses then you can mention that.

Answer:

We agree that it is importance to stress the novelty of this study on Swedish Warmblood, and thus we have added the following sentence in the text (L71 to L 76).

Also in conclusion can you mention any candidate gene or non-coding region want to study further?

Answer:

This is a very good question, actually we would like to study more on the hypermobility side in NS and on neurological – reactivity, instinctive reactions in SJ. We added a more general description on what we would like to do next at L527 to L531 in the conclusion.

In reference section 49 and 90 are same reference. Also in 90 the title of the paper written twice. 

Answer:

Thank you, they are now correct.

Round 2

Reviewer 1 Report

With regard to your second version of the manuscript «Genomic divergence in Swedish Warmblood horses selected for equestrian disciplines» of Ablondi et al., my overall recommendation is that the manuscript can be accepted in present form. Your cover letter answers all the questions, comments and suggestions I did in my first review. Overall, you have adequately corrected or improved your first versión. At most, I would recomend you a little change in your point of view of reflecting one of the results, as I stated in my previous review. I know that is a subjective concern, but I have considered to must be indicated.

Concretely, in Line 218-219, you still mantain the BIC analysis suggests two main subpopulations as the most likely number of subpopulations according to the figure 4a. In my opinion, the best figure to argue such sentence is the figure 4b. Anyway, I see you have broaden this topic in Discussion, but I consider you still mantain a subjective interpretation of a result in these lines, when there is no need to do it in this part, because you have the chance of discusse in the Discussion part.

Author Response

We have now included more information in the result section, clarifying the BIC graph in Figure 4a. We also explain that Figure 4b support the subdivision into two main subpopulations of SWB horses. (line 219-220, 343-344).

Reviewer 2 Report

Review Genes 612978 – Round 2

The authors from “Genomic divergence in Swedish Warmblood horses selected for equestrian disciplines” addressed all the comments from the last review round. I would like to add some minor comments about the answers from the last round. However, in my opinion, the manuscript is suitable for publication.

Authors’ answer

““the p-values used for the enrichment analysis, was performed any multi-testing correction?”

We agreed that a better explanation of the p-values presented in the GO term analyses is needed. As far as we know, it is very difficult to find a consensus on how to perform Panther analysis and it is still an ongoing trivial debate. Generally, as far as we understood, due to their structure, the GO terms, are not well suited for a multiple testing correction. (GO terms are highly dependent - terms on the same branch + one gene is annotated in several terms). Thus, we decided not to implement any multiple testing corrections. We clarified “Raw p-value” in the manuscript.

It is important to highlight that even with a highly dependent structure, it is possible to perform some types of enrichment analyses that remove the redundancy between the terms. Consequently, breaking this dependency between terms in the majority of the cases.

Authors’ answer

““several terms enriched with only one related gene” What this mean? Is this a real enrichment?”

A specific GO term could have one single gene assigned. This means that there are most probably no other genes in our study that do have the same specific gene ontology, but such genes could as well be included in several GO terms.

The question regards the cases were a single annotated gene is assigned to a term and the enrichment status of this term. In the case of the present study, the enrichment status is not the main guidance for the candidate gene selection. However, in studies such as RNA-sequencing. The use of these kind terms (with only one gene assigned) can cause “noise” in the functional analysis due to a “false-positive” enrichment cause in the majority of the cases by a lack of proper annotation for this tern.

Authors’ answer

To get Fst values that were more comparable to our XPEHH results, we analyzed the Fst divergence by sliding windows, instead of by single SNP Fst. 

We are absolutely aware of that the majority of the SNPs in the chip are neutral mutations without any functional roles but the genes in or close to where they are located in the detected genomic regions, might have a function.

The authors are correct in their answer. However, it is necessary to let as clear as possible in lines 365-366 that the possible functional effect it is not caused by the variants present in the SNP-chip (and consequently, evaluated in function of the Fst). The readers must be sure that the potential functional effects are caused by the causal variants in LD with the evaluated variants.

Author Response

““the p-values used for the enrichment analysis, was performed any multi-testing correction?”

We agreed that a better explanation of the p-values presented in the GO term analyses is needed. As far as we know, it is very difficult to find a consensus on how to perform Panther analysis and it is still an ongoing trivial debate. Generally, as far as we understood, due to their structure, the GO terms, are not well suited for a multiple testing correction. (GO terms are highly dependent - terms on the same branch + one gene is annotated in several terms). Thus, we decided not to implement any multiple testing corrections. We clarified “Raw p-value” in the manuscript.

It is important to highlight that even with a highly dependent structure, it is possible to perform some types of enrichment analyses that remove the redundancy between the terms. Consequently, breaking this dependency between terms in the majority of the cases.

Answer to the reviewer (round 2):

Thank you for pointing this out. Since the GO terms are mainly used for guidance among all genes present in the putative selected regions, we think the raw values will serve their function in this case. See also our reply below.

““several terms enriched with only one related gene” What this mean? Is this a real enrichment?”

A specific GO term could have one single gene assigned. This means that there are most probably no other genes in our study that do have the same specific gene ontology, but such genes could as well be included in several GO terms.

The question regards the cases were a single annotated gene is assigned to a term and the enrichment status of this term. In the case of the present study, the enrichment status is not the main guidance for the candidate gene selection. However, in studies such as RNA-sequencing. The use of these kind terms (with only one gene assigned) can cause “noise” in the functional analysis due to a “false-positive” enrichment cause in the majority of the cases by a lack of proper annotation for this tern.

Answer to the reviewer (round 2):

We will for sure take other methods of multi-testing correction into account in future RNA-seq studies. See also our reply above.

To get Fst values that were more comparable to our XPEHH results, we analyzed the Fst divergence by sliding windows, instead of by single SNP Fst. 

We are absolutely aware of that the majority of the SNPs in the chip are neutral mutations without any functional roles but the genes in or close to where they are located in the detected genomic regions, might have a function.

The authors are correct in their answer. However, it is necessary to let as clear as possible in lines 365-366 that the possible functional effect it is not caused by the variants present in the SNP-chip (and consequently, evaluated in function of the Fst). The readers must be sure that the potential functional effects are caused by the causal variants in LD with the evaluated variants.

Answer to the reviewer (round 2):

We have now further clarified this in the discussion (Line 370-371).